# Analyzing green and sustainable land use in China's coal cities: Insights from industrial transformation

**Hongfeng Zhang[1], Yinuo Zhao**[1]***, Fangteng Yang[2]**

**1** School of Public Administration and Policy, Shandong University of Finance and Economics, Jinan, China,
**2** School of Public Administration, Sichuan University, Chengdu, China

\* yn212109042@mail.sdufe.edu.cn

## Abstract

Driven by the goal of achieving sustainable development and carbon neutrality. Addressing environmental pollution and remediating land damage have become critical challenges in resource-based cities and regions with low land use efficiency. As a response, this study focuses on the 23 provinces where China's coal resource-based cities are situated. Utilizing data from 2014 to 2020, this research employs the SBM-Undesirable model, which considers undesirable outputs in efficiency calculations, and the Tobit regression test. It aims to explore the spatio-temporal variations in industrial transformation within resource-based cities and its impact on the efficiency of green space utilization. Furthermore, it analyzes the characteristics and the extent of the influence of factors such as industrial structure adjustments on urban land use efficiency, maximizing the output of land as a factor of production. The results show that: (1) Over the 7-year period studied, China consistently made nationwide adjustments to land area and land use structure to meet the needs of urban development (2) The regression test results show that the industrial transformation of resource-based cities can promote the improvement of green space utilization efficiency. The positive influence coefficient is 0.064 and is significant at a 1% level. (3) Environmental regulation, government expenditure, international trade, and green cover play a positive role in promoting green land use. The study provides valuable insights for policymakers and urban planners seeking to foster sustainable development in resource-based cities.

## Introduction

Promoting the green transformation and development of resource-based cities is an important way to save natural resources and an important measure to improve the level of urban modernization development. This transformation involves crucial actions such as restructuring industries and optimizing the allocation of production input factors, not only in China but also globally. Besides, industrial transformation and upgrading entail appropriate actions such as restructuring industries and adjusting production input factors allocation to achieve the transformation and upgrade of industries in resource-based cities in China and other countries

**Data Availability Statement:** The data underlying the results presented in the study are available from http://data.cnki.net/ https://www.nea.gov.cn/ http://www.stats.gov.cn/ https://www.mnr.gov.cn/ https://data.cnki.net/yearBook/single?nav=%E7%

BB%9F%E8%AE%A1%E5%B9%B4%E9%89%
B4&id=N2023050117 https://data.cnki.net/
yearBook/single?nav=%E7%BB%9F%E8%AE%
A1%E5%B9%B4%E9%89%B4&id=N2024030075
https://data.cnki.net/yearBook/single?nav=%E7%
BB%9F%E8%AE%A1%E5%B9%B4%E9%89%
B4&id=N2023110027 https://data.cnki.net/
yearBook/single?nav=%E7%BB%9F%E8%AE%
A1%E5%B9%B4%E9%89%B4&id=N2023110026
https://data.cnki.net/yearBook/single?nav=%E7%
BB%9F%E8%AE%A1%E5%B9%B4%E9%89%
B4&id=N2023010189 https://data.cnki.net/
yearBook/single?nav=%E7%BB%9F%E8%AE%
A1%E5%B9%B4%E9%89%B4&id=N2023010175
https://data.cnki.net/yearBook/single?nav=%E7%
BB%9F%E8%AE%A1%E5%B9%B4%E9%89%
B4&id=N2024010003 https://data.cnki.net/
yearBook/single?nav=%E7%BB%9F%E8%AE%
A1%E5%B9%B4%E9%89%B4&id=N2024010013
https://data.cnki.net/yearBook/single?nav=%E7%
BB%9F%E8%AE%A1%E5%B9%B4%E9%89%
B4&id=N2024010032 https://data.cnki.net/
yearBook/single?nav=%E7%BB%9F%E8%AE%
A1%E5%B9%B4%E9%89%B4&id=N2023030107
https://data.cnki.net/yearBook/single?nav=%E7%
BB%9F%E8%AE%A1%E5%B9%B4%E9%89%
B4&id=N2024030083 https://data.cnki.net/
yearBook/single?nav=%E7%BB%9F%E8%AE%
A1%E5%B9%B4%E9%89%B4&id=N2023030147
https://data.cnki.net/yearBook/single?nav=%E7%
BB%9F%E8%AE%A1%E5%B9%B4%E9%89%
B4&id=N2023030133 https://data.cnki.net/
yearBook/single?nav=%E7%BB%9F%E8%AE%
A1%E5%B9%B4%E9%89%B4&id=N2024010019
https://data.cnki.net/yearBook/single?nav=%E7%
BB%9F%E8%AE%A1%E5%B9%B4%E9%89%
B4&id=N2023010107 https://data.cnki.net/
yearBook/single?nav=%E7%BB%9F%E8%AE%
A1%E5%B9%B4%E9%89%B4&id=N2023030091
https://data.cnki.net/yearBook/single?nav=%E7%
BB%9F%E8%AE%A1%E5%B9%B4%E9%89%
B4&id=N2023050101 https://data.cnki.net/
yearBook/single?nav=%E7%BB%9F%E8%AE%
A1%E5%B9%B4%E9%89%B4&id=N2024030085
https://data.cnki.net/yearBook/single?nav=%E7%
BB%9F%E8%AE%A1%E5%B9%B4%E9%89%
B4&id=N2023080354 https://data.cnki.net/
yearBook/single?nav=%E7%BB%9F%E8%AE%
A1%E5%B9%B4%E9%89%B4&id=N2023110010
https://data.cnki.net/yearBook/single?nav=%E7%
BB%9F%E8%AE%A1%E5%B9%B4%E9%89%
B4&id=N2023030100 https://data.cnki.net/
yearBook/single?nav=%E7%BB%9F%E8%AE%
A1%E5%B9%B4%E9%89%B4&id=N2023050100
https://data.stats.gov.cn/easyquery.htm?cn=C01
https://gi.mnr.gov.cn/202402/t20240229_

[1]. This kind of action is of great significance for resource-based cities in China and other countries. Additionally, land productivity serves as an indispensable economic asset that plays a vital role in the transformation and upgrade of industries. Additionally, regulating the spatial input structure and quantity of land factors can provide security for industrial transformation and upgrading by harnessing the production capacity of land, which acts as one of the driving forces behind economic development [2]. Cultivating innovative industries is crucial for achieving urban transformation. Furthermore, nurturing emerging strategic industries has become a focal point in China's efforts to accelerate its journey toward becoming a strong manufacturing nation with high-quality standards. The process of transforming and upgrading industries while improving land use efficiency requires resource-dependent cities to overcome limitations associated with traditional energy-driven economies. Simultaneously, developing emerging industries can promote environmentally friendly land utilization practices, thereby unlocking comprehensive benefits derived from efficient land usage. As significant energy and resource supply bases during China's early years, resource-based cities made substantial contributions to national industrialization while serving as solid backbones supporting rapid economic growth at that time [3]. There are 262 resource-based cities in China. There are 262 resource-based cities in China, accounting for 40% of the total number of cities in the country (China's State Council released the National Sustainable Development Plan for Resource-based Cities (2013–2020)) [4]. However, in the early development of resource-based cities, the urban economy has developed significantly, the hidden problems left behind in the early development of resource-based cities are gradually exposed. Due to the lack of awareness of environmental protection and sustainable development, as well as the over-exploitation and rough processing of resources, these cities suffered from secondary environmental problems such as high energy consumption and severe damage to the ecological environment due to many carbon emissions during the early development process, manifesting as the "resource curse effect" in some resource dependent areas [5]. It is urgent to explore the new driving force for the growth of re-source-based cities. At this crucial time when China's economic growth is transforming from a stage of high-speed development to a phase of high-quality and sustainable development, the industrial transformation of resource-based cities has become the key step in implementing this concept, and the land elements in China's provinces are actively playing their factor vitality and utilization value. On the one hand, the vital role of land transformation and utilization in promoting industrial upgrading and optimizing the driving force of economic development has become the right thing to do to promote industrial upgrading in resource-based cities, improve the efficiency of green land utilization and achieve coordinated regional development under the new development pattern. Many cities in China are realizing the impact of organic combination of greening on surface and soil quality, which reflects the country's efforts in urban heat waves and land use carbon neutrality [6]. Under the new development pattern must face at this stage, how to improve the green utilization efficiency of industrial land in resource-based cities and achieve the urban green transformation is an important issue that resource-based cities and even cities where in need of land spatial layout adjustment. In addition, an in-depth investigation of other factors affecting the industrial restructuring of resource-based cities and providing theoretical references for industrial transformation and upgrading is also the focus of active discussion in the academic community.

This research based on the current development situation of resource-based cities, studies have been conducted on the remediation and restoration of abandoned industrial and mining land in urban areas, as well as constructively proposed direction and trend prediction of the green transformation of resource-based cities [7, 8]. These studies have contributed to an increasingly enriched theoretical system for the transformation of resource-based cities. Due to the many types and wide scope of resource-based cities, the research workload is large.

2838490.html https://www.mnr.gov.cn/sj/sjfw/kc_19263/zgkczybg/.

**Funding:** This research was funded by The Key Project of National Social Science Foundation of China (grant number 19AJY014); Major Project of Shandong Key Research and Development Program (Soft Science) (grant number 2023RZA02013); Shandong Province Social Science Planning Research Project (grant number 22CZTJ23); Provincial Applied Research Project of Humanities and Social Sciences of Shandong Province(2022-YYGL-08). The funders had no role in study design, data collection and analysis, decision to publish, or preparation of the manuscript.

**Competing interests:** The authors have declared that no competing interests exist.

Therefore, further research is still needed. On one hand, coal resource-based cities that heavily rely on mining and processing ores, particularly coal which is a highly polluting resource, have experienced numerous environmental ecological pollution problems [9]. Therefore, this paper focuses on Chinese resource-based cities where coal plays a major role. On the other hand, few scholars have examined industrial transformation and upgrading from a comprehensive perspective by dynamically analyzing influencing factors related to industrial restructuring in resource-based cities while considering their relationship with land use. To address this knowledge gap, this paper will analyze the characteristics and degree of influence that urban land use and other factors exert on industrial restructuring using dynamic land use measurement combined with a non-expected output model. Additionally, it will explore factors influencing industrial restructuring and upgrading in resource-based cities through Tobit regression analysis while also examining how industrial restructuring promotes green urban land use under multiple concurrent factors, and the factors of constraints, through scientific ways to eliminate the negative impact of constraints. Based on these findings, theoretical guidance for promoting green and sustainable development in coal resource-based cities will be provided along with empirical conclusions regarding future directions for industrial structure upgrading and adjustment.

## Theoretical analysis framework

China has many resource-based cities scattered all over the country. The superior natural resource conditions have provided sufficient raw materials for industrial development and enabled China to have an independent and complete industrial system and a steadily developing national economy. However, the over-consumption of natural resources has led to environmental pollution and a lack of production incentives, which have seriously hindered the development of resource-based cities. In addition, the in-crease in uncertainty and instability in the general international political and economic environment has given rise to many unbalanced development problems, which also threaten the coherence and stability of China's economic development.

### Review of the literature

Resource-based cities are characterized by their heavy reliance on the exploitation and processing of natural resources, such as minerals and forests. Since the 1960s, these cities have witnessed a shift in energy sources, with traditional resources like coal facing increasing competition from newer, more energy-efficient alternatives such as oil, cleaner natural gas, and nuclear energy. Simultaneously, the tightening of environmental regulations has objectively imposed restrictions on traditional energy resources. This has resulted in the decline of numerous towns overly dependent on these traditional resources, leading to reduced overall economic efficiency and a rise in unemployment, prompting people to migrate elsewhere [10].

This transformation of resource-based cities has drawn significant attention from experts and scholars, both domestically and internationally. A primary focus has been on identifying new drivers for the economic development of these cities. Many resource-rich regions grapple with the so-called "resource curse" effect [11]. The resource curse implies that countries or regions with abundant natural resources often struggle to leverage this wealth for economic development, resulting in low levels of industrialization and difficulties in diversifying their economies due to over-reliance on a single resource. Some scholars have challenged the universality of the "resource curse" theory by demonstrating that not all resource-based regions fall victim to this phenomenon. Subsequently, researchers have delved into the institutional aspects of resource-based cities [12], raising questions about whether state institutions can

mitigate the resource curse through specific designs. Mehlum et al. [13], for instance, argue that the organization of governments is a pivotal factor influencing whether regions can mitigate the adverse effects of resource dependency.

In exploring the developmental dynamics and obstacles of resource-based cities under varying conditions, scholars, both in China and abroad, have conducted studies encompassing several elements of urban development. These investigations have revealed that policy preferences [14], location conditions [15], and the interests of key stakeholders [16] significantly influence the transformation of resource-based cities. In the context of China's resource-based cities, the most pressing task for sustainable development is promoting industrial transformation and accelerating changes in the economic development pattern. Over the past half-century, these cities have made significant contributions to China's economic growth by supplying essential raw materials for industrial production. However, rapid industrialization and urbanization have generated a substantial demand for resource extraction, particularly coal, which has led to the gradual depletion of mineral resources and inflicted severe damage on the ecological environment [17]. Shifting the economic development model and reshaping the industrial structure have thus become fundamental goals for China's industrial transformation and upgrading, essential for achieving sustainability.

According to the theory related to land use efficiency, land use efficiency refers to the efficiency level of economic, social and ecological benefits generated by the allocation and use of land resources in different regions and economic sectors [18]. Land use efficiency serves as a critical indicator of how effectively land functions as a production factor. When it comes to the efficiency of green land use, influencing factors primarily include physical geographical elements like land topography and the ecological environment, as well as economic development factors, especially environmental benefits and ecological value post-land utilization in market-driven conditions. Typically, this is measured by the ratio of green coverage per unit of land area to the total built-up area. Furthermore, social factors, including policies and regulations, also play a significant role. With socioeconomic development, innovation has emerged as a pivotal driver for sustainable land use [19]. Through the discussion of scholars, it is essential to recognize that land's spatial immutability necessitates that the adjustment of industrial structure considers the input factors required for production and the impact of land use on industrial agglomeration effects.

Land, as a fundamental production factor, holds a crucial role in urban transformation, and land use transformation serves as a reflection of evolving land use conflicts. As urbanization and industrialization advance, influenced by various external pressures and internal resistance, land use transformation extends beyond the scope of a single land type to encompass a diverse range of land types integrated into the urban landscape [20–22]. The transformation of land use types is no longer limited to single categories but rather involves a blend of various land types. This has generated significant attention from scholars, both nationally and internationally, as they seek to understand how socioeconomic factors impact the challenges related to land use transformation [23, 24]. These factors not only exert an endogenous influence on location choices but also contribute to the objective adjustment of urban land use patterns. Scholars like Long Hualou et al. [25] have introduced the concept of land-use transformation in China, defining it as a process that evolves over a period aligned with economic and social development stages. At the same time, some scholars pay attention to the impact of the organic combination of urban greening on the surface and soil quality, which reflects China's efforts in urban heat wave and carbon neutrality in land use. This process involves changes in various attributes, such as land type, land function, and land property rights. As China enters the middle and late stages of industrialization, the emphasis on environmental protection and the transformation of the industrial structure has led to the relocation of many industrial

enterprises from urban centers, leaving behind considerable amounts of vacant or abandoned industrial land. This has become a substantial component of land use transformation [26–28].

The structure of the industrial sector is central to driving economic growth, and any country or region seeking sustained economic progress must optimize and upgrade this structure. The industrial structure represents the composition of various economic sectors, along with their interconnections and proportions [29]. Upgrading the industrial structure entails a shift from lower-level, lower-value-added industries to higher-level, higher-value-added sectors [8]. This optimization process leads to the migration of high-emission enterprises. Thus, the relationship between land production factors and the adjustment of the industrial structure is inherently intertwined [30]. Land, as a critical production factor, is intrinsically linked to industrial restructuring. In line with these theories, the optimization and upgrading of the industrial structure have the potential to influence the input ratios and layout structure of urban land resource elements. This transformation also leads to changes in land use types, thereby regulating land output efficiency. The linkage between industrial restructuring and green land utilization efficiency is complex, and the increasing cross-regional flow of resources. Constrained by urban environmental protection policies and behavioral ethics, the Chinese government, under the influence of the goal of achieving sustainable development [31], inevitably place constraints on the industrial restructuring and green land-use efficiency of resource-based cities, influenced by ecological protection and related policies.

At this stage, the transformation of the secondary industry among the three major industrial structures is key to the development of resource-based cities. Promoting the transformation and upgrading of the secondary industry can unlock the economic potential of land, contributing to the coordinated development of all three major industries. Several aspects have been discussed by scholars. For instance, studies on price mechanisms have primarily examined how industrial land prices impact the productivity of industrial enterprises, revealing that distortions in local government industrial land prices can significantly reduce industrial productivity [32, 33]. Research into the influence of the extent of industrial land expansion on urban innovation suggests that land resource allocation negatively affects urban innovation [34, 35]. At the same time, after the government introduced policies related to housing construction restrictions, China's urban land allocation will largely give more space to industrial land for a considerable period of time [36]. From the perspective of local governments' differential allocation of industrial and service land, it has been found that while land finance contributes to industrial development, it hinders the growth of the tertiary sector, particularly the service sector, from the standpoint of land use structure [37].

As the social economy develops, environmental pollution resulting from emissions in the process of urban industrial development has become increasingly prominent. As can be seen from Table 1, the energy consumption of coal resource-based areas is heavily dependent on coal, and the clean degree of energy consumption structure is not high. The coal resource-based cities mainly use coal as energy, which indicates that the coal resource-based cities have obvious characteristics of self-sufficiency. Scholars have begun to explore pathways for optimizing the industrial structure under environmental constraints, considering the rising importance of sustainable development and green practices [38, 39]. In summary, the relationship between green urban land use, the low-carbon economy, and ecological construction has gained traction. While these studies have yielded varying conclusions, they collectively underscore the idea that the industrial transformation and upgrading of resource-based cities are multifaceted and influenced by an array of factors. However, in order to completely realize urban transformation and sustainable economic development, it is necessary to further understand the influencing factors of green land use efficiency and their mechanism of action. According to the detailed analysis of the influencing factors, targeted exploration of solutions.

**Table 1. Energy consumption structure in coal-based areas (%).**

|  | Coal | Oil | Natural gas | Electricity |
|---|---|---|---|---|
| 2014 | 70.6 | 15.2 | 8.3 | 5.9 |
| 2015 | 67.4 | 17.7 | 8.7 | 6.1 |
| 2016 | 74.2 | 12.6 | 6.4 | 5.8 |
| 2017 | 75.1 | 12.1 | 6.9 | 5.9 |
| 2018 | 66.9 | 17.5 | 9.3 | 6.3 |
| 2019 | 67.8 | 17.2 | 8.6 | 6.3 |
| 2020 | 68.6 | 16.5 | 7.5 | 6.5 |

Source: National Energy Statistical Yearbook

https://www.nea.gov.cn/

In this way, it can not only help to improve the green use of land in resource-based cities, but also provide theoretical reference for the subsequent research on the promotion path.

## Influence mechanism of industrial restructuring in resource-based cities

The transformation of resource-based cities is progressing at a rapid pace, and it has prompted adjustments in the industrial structure. The industry transformation of resource-based city also has achieved some success, the GDP of China's 262 resource-based urban areas will increase from 15.7 trillion yuan to 26.8 trillion yuan in 2021, with an average annual growth rate of 6%. Land, as a vital factor of economic development [40], is profoundly influenced by these transformations, leading to variations in land usage and output efficiency across different industrial layouts [41]. China has long grappled with the challenge of an imbalance between its population and available land resources. Presently, China's land resources are insufficient to meet the demands of urbanization, hampering the realization of their full economic, social, and ecological potential.

The process of upgrading the industrial structure, through the fine-tuning of input ratios and output efficiency of resource factors, serves a dual purpose. Firstly, it promotes dynamic enhancements in the efficiency of green land utilization. Secondly, it triggers changes in land usage with economic, social, and ecological implications, the process is shown in Fig 1.

Economically, the industrial transformation in some resource-based cities plays a vital role in optimizing and adjusting the proportion of the three major industries relative to their respective locations. The upgrading of the industrial structure directly enhances the economic output efficiency of land utilization in cities through the expansion of industrial value chains and the deep processing of products [42]. Notably, resource-based cities have historically relied heavily on mineral enterprises [43]. This over-reliance has become a major hindrance to the development quality of these cities due to the prominence of inefficient and highly polluting industries [44]. By upgrading the industrial structure, resource-based cities can mitigate their dependence on land resources. This transformation is reflected in the infusion of innovative technologies into traditional industries and the shift in economic development paradigms, facilitating a transition from inefficient and extensive land use to a more intensive and efficient green utilization model. Additionally, industrial restructuring leads to changes in the output levels of the three major industries, unlocking higher economic benefits from land as a production input factor [45].

From a social perspective, industrial restructuring involves the transformation of traditional industries, the elimination of outdated sectors, and the cultivation of new industries, which leads to the optimization and upgrading of the industrial structure. Industrial transformation

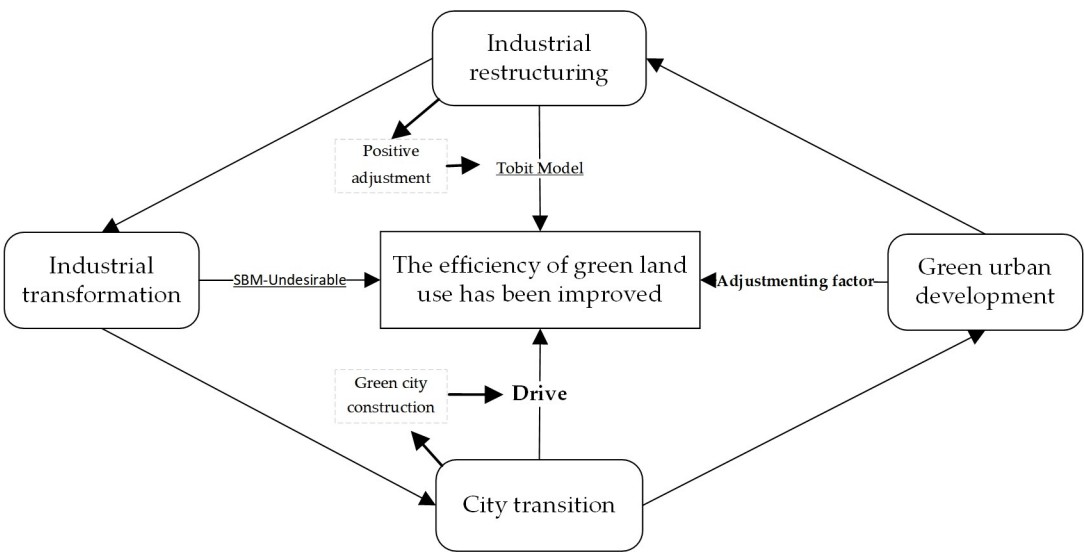

**Fig 1. The conduction of industrial transformation to land green utilization.**

and upgrading have a ripple effect on the optimization of the urban land usage structure and labor employment proportions [46]. This, in turn, fine-tunes the allocation of factors such as land, labor, capital, and technology, enhancing the comprehensive efficiency of urban land resources [47]. Escalating land costs and the pursuit of profit growth drive enterprises to adjust the ratio of various factor inputs, seeking to replace land with capital, human resources, and technology, thereby reducing their reliance on land [48, 49]. This not only reduces enterprise expenses but also conserves inputs while bolstering the innovation capabilities and levels of these enterprises, further promoting green land utilization. Moreover, as the industrial structure is upgraded, factor inputs are adjusted accordingly, with the most tangible change being land consolidation. Given the immutability of land's spatial location, transformed industries generate demand for new land, while previously used land vacated or abandoned by departing traditional industries necessitates renewal and adjustment. This leads to the re-planning and integration of land, a process that coincides with the movement of land attachments and other elements. In some resource-based cities, these developments may result in the hollowing out of industrial land and the emigration of urban populations [50]. Consequently, various aspects of society are optimized through the adjustment of the industrial structure.

From an ecological standpoint, the transformation of the industrial structure diminishes the negative externalities of environmental pollution by enhancing "economic and ecological serviceability." It elevates the ecological benefits of urban land allocation [51]. Traditional industries in resource-based cities are often characterized by high energy consumption and pollution levels [52], leading to issues such as atmospheric pollution, resource depletion, and biodiversity loss. These problems are detrimental to the sustainable development of resource-based cities. The upgrade of the industrial structure serves a dual purpose. Firstly, it promotes the extension of the original resource-dependent industrial chain, nurturing new green industries and stimulating diversified economic growth in resource-based cities. Secondly, regarding green land use, industrial restructuring facilitates an increase in environmental protection land and ecological construction areas, gradually reducing environmental losses and unlocking the "industrial structure dividend." This, in turn, enhances the ecological and environmental benefits of urban land. The coordination of industrial restructuring and green land utilization

reflects several objectives, including optimizing resource efficiency, minimizing environmental pressures, and fostering synergy between ecology and the economy [53].

## Materials and data

This paper focuses on the impact of industrial restructuring on land use efficiency in coal resource-based cities. Most of the 262 resource-based cities in China are facing the problems of resource depletion, industrial homogeneity, economic decline, lack of subsequent development momentum, and severe damage to the ecological environment. To further study the impact of industrial structure adjustment on green use of land efficiency, this study first adopts the dynamic land use attitude model to measure the degree of dynamic changes in urban land, to quantitatively analyze the rate of change of industrial and mining land in coal resource-based cities from 2014 to 2020, and thus define the changes in the quantity of land use affected by the industrial restructuring. Secondly, the "input-desirable-undesirable" framework is constructed based on the SBM-Undesirable Output Model to calculate the spatial and temporal patterns of industrial restructuring and green land use efficiency changes in resource-based cities. Finally, the Tobit regression model is used to verify the robustness of the factors influencing industrial restructuring in resource-based cities based on the influence of industrial restructuring on green use of land efficiency, in conjunction with land use changes. Specific research methods are as follows:

### Land use dynamic model

The land use dynamic model can quantitatively reflect the change rate of regional land use quantity and predict the future trend of land use change and is one of the frequently-used analytical methods used in land use and planning studies [54]. The land uses dynamic model is significant to the traditional quantitative analysis model. The model takes into consideration the quantitative and spatial attributes of the transformation of different land use types and can measure and compare the overall or integrated dynamic degree of regional land use change. A positive land use dynamics result means that the land type has changed dramatically and increased in quantity over the period, while a negative number means that the land type has changed steadily and the amount is decreasing over the period. The single land-use dynamic reflects the rate of change in the area of a land-use type in the study area over a given time frame, and the model focuses on analyzing the intensity of change in each land-use type.

This is example 1 of an equation:

$$K = \left[\left(U_i - U_j\right)/U_i\right] \times \frac{1}{T} \times 100\% \tag{1}$$

where K is the degree of change in the dynamics of the land area; $U_i$ is the relative initial time of the study land class as a percentage of the total area; $U_j$ is the relative end time of the study land area as a percentage of the total area; T is the time interval.

### SBM-Undesirable model

The SBM-Undesirable model is a derivative of the traditional DEA model and can effectively solve the problem of measuring efficiency with slack variables and undesired outputs [55]. The SBM Undesirable model is based on the traditional DEA model. The indicators that can measure the efficiency of land use are selected from relevant studies, as shown in Table 2.

To better explain the industrial restructuring industrial transformation and upgrading as the core influencing factors (variables) of land use efficiency, this paper selects the explanatory variables in Table 3 as land use efficiency in resource-based cities, the core explanatory

**Table 2. Variables associated with non-expected output models.**

| Variable type | Variable name | Definition |
|---|---|---|
| Input variables | Land elements | Percentage of building land area [61] |
| | Capital elements | Fixed asset stock |
| | Government input | Local government financial expenditure [65] |
| Expected output | Economic benefits | GDP generated from construction land in municipal areas [61] |
| | Ecological benefits | Greenery coverage in built-up areas [56] |
| Unwanted output | Environmental pollution | Exhaust emissions ($SO^2$) [63] |

variables as industrial restructuring and the control variables as economic development level, financial support, population density, and investment level.

## Tobit regression model

Ordinary least squares (OLS) regression is prone to biased results, but the Tobit model can solve the problem of a 'limited dependent variable'. The estimated value of industrial structure adjustment in resource-based cities is greater than 0, which is a "limited dependent variable", so this paper adopts a general regression method and introduces control variables to analyze land use efficiency and industrial transformation and upgrading.

This is example 2 of an equation:

$$Y_{it} = \beta_0 + \beta_i X_{it} + \varepsilon_{it} \tag{2}$$

where, $Y_{it}$ represents the value of the city's industry in year t of the city i, $X_{it}$ is its influence factor, $\varepsilon_{it}$ denotes the random error term.

The relationship between the above methods and the research topic is shown in Fig 1.

## Description of variables

The outcome variable of this study is the green land use efficiency, which is the ratio of the green output of land in built-up areas to the input factors such as labor, capital, and energy consumed. In addition, based on the constructed "input-desirable-undesirable" framework, data related to industrial structure, investment level, land construction, environmental regulation, government expenditure, population density, international investment, and land green cover will be used as explanatory variables for the effect of industrial structure adjustment on land use efficiency.

## Green cover

The evaluation of land use efficiency should focus on the overall benefits of urban land and, most importantly, its environmental benefits [56]. The green land use efficiency calculated

**Table 3. Explanatory variables associated with the non-expected output model.**

| Explanatory variables | Variable name | Definition |
|---|---|---|
| Explained variables | Green land use efficiency | Based on SBM-Undesirable measurements (the result of Table 2) |
| Ore explanatory variables | Industrial transformation | Based on industry structure level measurement [58] |
| Control variables | Level of economic development | GDP per capita in urban areas |
| | Urban population density | Ratio of the number of permanent residents to the area of the built-up area in the municipality |
| | Financial support | Local government general financial expenditure |
| | Investment level | Amount of local fixed asset investment |

based on input-output factors such as green land coverage area and public service construction can better explain the green use of urban land resources at the level of optimal allocation and intensive use. Green cover of urban land can dilute the pollution degree of industrial production emissions to air quality and natural environment [57], and improve the coordination between new-type urbanization and land green use efficiency to a certain extent, to realize the optimal allocation of urban land resources and land green use.

## Industrial structure

Population numbers and policy preferences drive industrial development, and the further expansion of industrial activity can lead to population shifts, causing a siphon effect and economies of scale in the industry [58]. The adjustment and upgrading of the industrial scale drive urban economic growth, which means many local governments cannot but carefully consider the quantity and direction of construction land layout. From a dynamic perspective, the industrial structure adjusting is a coordinated and interactive process between the rationalization of the industrial structure and the industrial structure updating [59]. This paper uses the ratio of the gross value of the secondary industry and the tertiary industries to express the industrial structure.

## Investment levels

Consumption, investment, and exports are widely regarded as vital drivers of economic growth [60]. Among them, investment plays an essential role in capital formation. Industrial transformation requires a large amount of capital input to help it play its sufficient part, which requires not only speeding up the flow of factors but also harmonizing the industrial structure between regions. In this research, the natural logarithm of the total social investment in fixed assets is chosen as representing the investment in fixed assets.

## Land building

Land construction is the core explanatory variable of this paper. As the size of the urban population has accumulated over time, economic development and the optimization and upgrading of the industrial structure are a necessity for socio-economic development [61]. This forces local governments to adjust the land supply scale and the layout of land use planning. According to the existing literature, the urbanization level is generally measured by the land scale of built-up areas, so this paper uses the proportion of urban construction land in urban land to represent the urbanization degree of land [62].

## Environmental regulation

Environmental protection is increasingly becoming the first consideration for government action, and investment in environmental protection can also improve the industrial structure and infrastructure in the region [63]. The environmental protection industry is a crucial driver of China's economic growth. The consumption of resources is the driving force behind China's economic growth and is a principal driver of secondary industry development [64]. At the same time, however, the consumption of traditional energy sources is also a main source of pollution. There is currently no unified academic indicator for measuring the intensity of environmental regulation, and given the availability of data, environmental pollution indicators are based upon a comprehensive calculation of industrial emissions based on the entropy weighting method. Here we choose the natural logarithm of the annual amount of $SO^2$ emitted

at the local level as the measured degree of environmental protection regulation. The size of the numerical value corresponds to the laxity or strictness of environmental regulations.

## Government spending

Economic development relies on a rational industrial structure and government policies. Before all, regional industrial policy plays a powerful role in industrial structures. The fiscal revenue is lean toward high-tech and clean enterprises, and government spending is adjusted to guide structural changes in market supply and demand, it also promotes optimal resource allocation. One of the more commonly used indicators in most studies to proxy for government size and fiscal auto-stabilizers is the ratio of general government expenditure to GDP [65]. Hence, the government expenditure in this paper is expressed by the proportion of local budget expenditure in the current year's GDP.

FDI plays a significant role in increasing capital stock, improving national trade, raising technology levels, and accelerating industrial upgrading has a direct bearing on the level of industrial upgrading as well [66, 67]. On the contrary, foreign investment has a direct impact on the level of industrial upgrading. In addition, foreign investors are attracted by China's low-cost environment and labor force, and the expansion of regional FDI concentration scale or GDP growth occurred. Therefore, this paper selects the actual utilization of FDI to represent international investment and treats it as a natural logarithm to eliminate the effect of heteroskedasticity.

## Data sources

2021 China accelerates the construction of demonstration zones to accelerate the green transformation of industries and the sustainable use of various types of land, which means that the industrial transformation of resource-based cities enters a new stage. To further inspect the results of industrial structure transformation and the standard of urban green land use in resource-based cities of China, this study selects 23 provinces where 68 coal resource-based cities are located out of 262 resource-based cities defined by the State Council of China in 2014 as the research sample and setting the period from 2014 to 2020, exploring the influence of industrial structure transformation on urban land use. This study will discuss and investigate the impact of industrial restructuring on green land use in Chinese coal resource cities.

The land image data in this paper is from the remote sensing monitoring data of land use from 2014 to 2020 of the Data Centre for Resource and Environmental Sciences, Chinese Academy of Sciences1. According to its classification system, construction land includes urban land, rural settlements, industrial and mining land in urban villages, and so on. The socio-economic panel data are from the China Urban Statistical Yearbook and China Statistical Yearbook, 2014–2021.

## Empirical analysis

### Analysis of spatial and temporal changes in land use in resource- based cities

As can be seen from Fig 2, the overall urban construction land in China's 23 provinces has changed in varying degrees from 2014 to 2020 show in Table 4. The construction land in China's resource-based cities from 2014 to 2020 has changed a lot, with the extreme difference in land use dynamics reaching 5.3, indicating that China's construction land planning and utilization are constantly adjusting over the seven years.

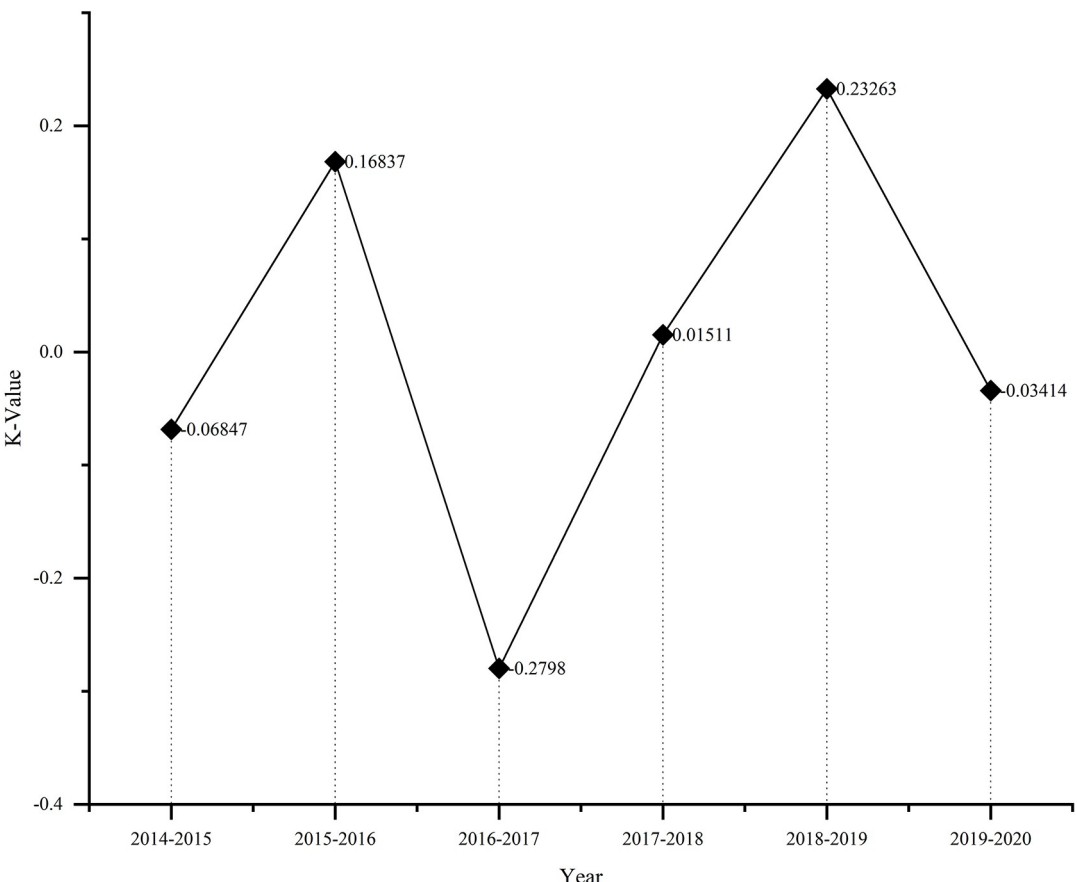

**Fig 2. Change in land use dynamics in resource-based cities.** http://www.stats.gov.cn/.

From the perspective of time, the land use dynamic attitude in 2016–2017 was -0.28, which indicates that the National Land Reclamation Plan (2016–2020) (hereinafter referred to as the Plan), which was formulated in 2015, put forward strict safeguard measures the optimization of the reclamation level and layout structure of construction land, effectively controls the area of construction land, and improves the integrated carrying capacity of the land. The dynamic attitude of construction land in 2018–2019 is 0.23, indicating that the area of construction land in resource-based cities has increased during this period, and urban abandoned industrial and mining land has been remediated and reused through scientific means, which has increased the amount of construction land available in resource-based cities and improved land use efficiency.

From the perspective of time, the land use dynamic attitude in 2016–2017 was -0.28, which indicates that the National Land Reclamation Plan (2016–2020) (hereinafter referred to as the Plan), which was formulated in 2015, put forward strict safeguard measures the optimization of the reclamation level and layout structure of construction land, effectively controls the area of construction land, and improves the integrated carrying capacity of the land. The dynamic attitude of construction land in 2018–2019 is 0.23, indicating that the area of construction land in resource-based cities has increased during this period, and urban abandoned industrial and mining land has been remediated and reused through scientific means, which has increased the amount of construction land available in resource-based cities and improved land use efficiency.

**Table 4. Land use dynamics in provinces where resource-based cities are located.**

| Province | K-Value (year) | | | | | |
|---|---|---|---|---|---|---|
| | 2014–2015 | 2015–2016 | 2016–2017 | 2017–2018 | 2018–2019 | 2019–2020 |
| Ningxia Hui Autonomous Region | 0.059 | 0.059 | 0.017 | 0.060 | -0.491 | 0.934 |
| Gansu Province | 0.058 | 0.025 | -0.083 | -0.022 | 0.011 | -0.029 |
| Shaanxi Province | 0.022 | 0.047 | 0.111 | -0.003 | 0.086 | -0.587 |
| Shanxi Province | 0.047 | -0.377 | 0.862 | -0.127 | -0.078 | 0.052 |
| Yunnan Province | 0.020 | 0.041 | 0.016 | 0.062 | 0.008 | 0.017 |
| Guizhou Province | -0.366 | 0.115 | 0.743 | -0.085 | 0.415 | -0.533 |
| Sichuan Province | 0.045 | 0.135 | 0.054 | 0.079 | 0.016 | 0.020 |
| Chongqing | 0.085 | 0.057 | 0.028 | 0.049 | 0.037 | 0.102 |
| Hainan Province | 0.447 | -0.060 | 0.896 | -0.630 | 0.085 | -0.495 |
| Guangxi Province | 0.077 | 0.053 | 0.058 | 0.033 | 0.018 | 0.042 |
| Guangdong Province | 0.000 | -0.003 | -0.222 | -0.109 | 0.236 | -0.180 |
| Hunan Province | -0.010 | 0.042 | 0.125 | 0.012 | -0.032 | 0.110 |
| Hubei Province | -0.236 | 0.042 | 0.225 | 0.039 | -0.462 | -0.005 |
| Henan Province | 0.093 | 0.049 | 0.111 | 0.027 | -0.113 | -0.063 |
| Shandong Province | 0.018 | 0.065 | 0.034 | 0.058 | 0.058 | -0.011 |
| Jiangxi Province | 0.131 | -0.004 | 0.097 | 0.066 | -0.194 | 0.834 |
| Anhui Province | 0.057 | 0.019 | 0.021 | 0.024 | 0.016 | 0.083 |
| Jiangsu Province | 0.049 | 0.046 | 0.019 | 0.001 | 0.004 | 0.046 |
| Heilongjiang Province | -0.855 | 1.255 | -0.445 | -0.019 | -0.179 | 0.075 |
| Jilin Province | -0.003 | -0.013 | -0.001 | 0.016 | 0.019 | -0.429 |
| Hebei Province | 0.098 | -0.113 | 0.217 | -0.155 | 0.060 | 0.153 |
| Inner Mongolia Autonomous Region | 0.021 | -0.092 | 0.024 | 0.128 | -0.112 | 0.005 |
| Liaoning Province | 0.074 | -0.242 | 0.478 | 0.016 | -0.443 | -0.065 |

https://www.mnr.gov.cn/

From a spatial perspective,show in Fig 3, China is adjusting the land area and land use structure throughout the country. Specifically, Jiangxi Province, located in southern China, has the highest negative land use dynamics among the 23 provinces where resource-based cities are located, with a dynamic of -0.16, which indicates a contraction in land use change in Jiangxi Province, i.e., a contraction in the area of land used for construction in Jiangxi Province. Heilongjiang Province ranked highest in positive attitudes, with a positive attitude of 0.14. This value indicates that the industrial restructuring and upgrading of Heilongjiang Province has promoted the reclamation of abandoned industrial and mining land and the linkage of urban and rural construction land, which drives the consolidation and redevelopment of construction land in traditional industrial bases.

## Analysis of spatial and temporal changes in land use efficiency in resource-based cities

Based on the results obtained from the undesirable outputs model (Slacks-Based Measure; SBM-Undesirable), According to Tables 5 and 6, the green land use efficiency of various cities changes dynamically from 2014 to 2020, and three rank intervals were divided using the natural break point method of ArcGIS. In specific analysis, the economic efficiency of green land use in central and western regions and northern regions of China in 2014 is at the midstream level, especially in Inner Mongolia Autonomous Region, Shanxi Province, and Gansu

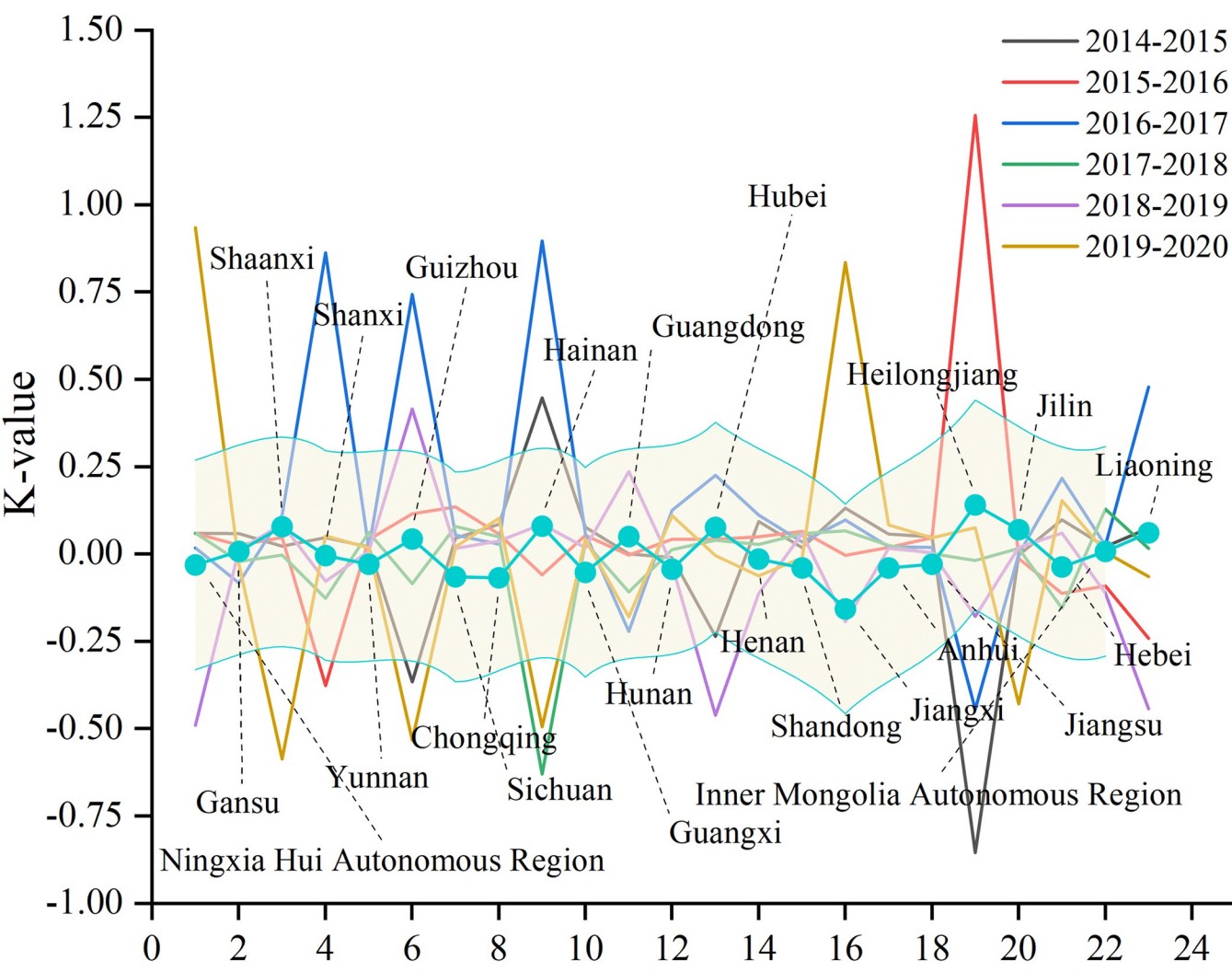

**Fig 3. Dynamic changes of land use in resource-based cities.** http://www.stats.gov.cn/.

Province. These provinces are typically resource-dependent regions, with an industrial structure dominated by the primary and the secondary industries, and a secondary coal mining industry that relies on primary processing for instance coal mining, with high energy consumption and low value added in the industrial chain. What is noticeable is that these provinces and municipalities did not undertake large-scale industrial transformation and upgrading in 2014, thus lagging other regions that have taken the lead in industrial transformation, resulting in their low level of economic development and slow development. 2016 saw a significant decline in green land use efficiency in Yunnan and Hainan provinces, and an increase in green land use efficiency in Guizhou and Hunan provinces, indicating that the industrial landscape in southern China is changing. Facts have proved that with rapid economic development and the changing social environment, re-source-dependent regions cannot sustain economic growth if they persist in mineral extraction and rough processing. Regions with a more backward economic development base should carry out industrial restructuring, such as taking advantage of their own natural environmental advantageous conditions to vigorously develop the tourist industry, extend the agricultural industry chain, carry

**Table 5. Change of green land use efficiency in coal resource-based cities.**

| Province | Land Utilization Efficiency | | | | | | |
|---|---|---|---|---|---|---|---|
| | **2014** | **2015** | **2016** | **2017** | **2018** | **2019** | **2020** |
| Ningxia | 0.1732 | 0.1632 | 0.1446 | 0.1401 | 0.1590 | 0.2548 | 0.1919 |
| Gansu | 0.3324 | 0.2759 | 0.2007 | 0.1752 | 0.2017 | 0.2471 | 0.2388 |
| Shaanxi | 0.4908 | 0.4566 | 0.3733 | 1.0000 | 1.0000 | 1.0000 | 1.0000 |
| Shanxi | 0.3146 | 0.2802 | 0.3145 | 0.3392 | 0.3680 | 0.3987 | 0.3658 |
| Yunnan | 1.0000 | 0.5746 | 0.4746 | 0.3260 | 0.2981 | 0.3675 | 0.3329 |
| Guizhou | 0.3967 | 1.0000 | 1.0000 | 0.2624 | 0.2634 | 0.2521 | 1.0000 |
| Sichuan | 0.4886 | 0.4682 | 0.3393 | 0.3177 | 0.3085 | 0.3321 | 0.2990 |
| Chongqing | 1.0000 | 1.0000 | 1.0000 | 1.0000 | 1.0000 | 1.0000 | 1.0000 |
| Hainan | 1.0000 | 1.0000 | 0.2705 | 0.1723 | 0.2879 | 0.3382 | 1.0000 |
| Guangxi | 0.4169 | 0.3932 | 0.3069 | 0.2510 | 0.2317 | 0.2704 | 0.2457 |
| Guangdong | 1.0000 | 1.0000 | 1.0000 | 1.0000 | 1.0000 | 1.0000 | 1.0000 |
| Hunan | 0.4899 | 0.6573 | 1.0000 | 0.4084 | 0.3642 | 0.4104 | 0.3666 |
| Hubei | 0.4261 | 0.5557 | 0.4465 | 0.3587 | 0.3564 | 1.0000 | 0.5602 |
| Henan | 0.5749 | 0.5006 | 0.4256 | 0.3903 | 0.3820 | 0.4475 | 0.4410 |
| Shandong | 0.5825 | 0.5717 | 0.4682 | 0.4287 | 0.4011 | 0.4434 | 0.4312 |
| Jiangxi | 0.4308 | 0.3863 | 0.3044 | 0.2532 | 0.2415 | 0.3183 | 0.2154 |
| Anhui | 0.4549 | 0.4247 | 0.3205 | 0.2826 | 0.2866 | 0.3083 | 0.2869 |
| Jiangsu | 1.0000 | 1.0000 | 1.0000 | 1.0000 | 1.0000 | 1.0000 | 1.0000 |
| Heilongjiang | 0.2485 | 0.3055 | 0.1575 | 0.1639 | 0.1707 | 0.1854 | 0.1762 |
| Jilin | 0.2060 | 0.2330 | 0.1843 | 0.1745 | 0.1621 | 0.1962 | 0.2519 |
| Hebei | 0.4173 | 0.3863 | 0.4715 | 0.2962 | 0.2936 | 0.3234 | 0.2821 |
| InnerMongolia | 0.2533 | 0.2487 | 0.2492 | 0.2207 | 0.2427 | 0.2993 | 0.2754 |
| Liaoning | 0.3370 | 0.3178 | 0.3390 | 0.2710 | 0.2732 | 0.3882 | 0.3627 |

out deep processing of crops, and innovate production methods. From 2018 to 2020, green land use efficiency in Sichuan Province, Anhui Province, and Guangxi Zhuang Autonomous Region declines sharply and at an inefficient level, which is related to their unclear direction of industrial structure transformation and short supply of production of raw materials. Industrial restructuring can hardly do without the support of the production of raw materials, and as provinces compete to embark on industrial upgrading, the stability and security of the industrial chain have propped up the raw materials and technologies required for industrial development. In addition, through the dynamic changes in green land use from 2014 to 2020, it is found that some regions have maintained a stable green land use efficiency level, such as Jiangsu Province, Guangdong Province, and Chongqing Municipality in the high-efficiency zone, and Inner Mongolia Autonomous Region, Heilongjiang Province, Jilin Province, and Gansu Province in the low-efficiency zone.

**Table 6. Natural break point interval.**

| Section | 2014 | 2015 | 2016 | 2017 | 2018 | 2019 | 2020 |
|---|---|---|---|---|---|---|---|
| Low Efficiency | 0.1732 -0.3369 | 0.1632 -0.3863 | 0.1446 -0.2705 | 0.1401 -0.2709 | 0.1589 -0.2634 | 0.1854 -0.3083 | 0.1762 -0.2989 |
| Middle Efficiency | 0.3369 -0.5825 | 0.3863 -0.6573 | 0.2705 -0.4746 | 0.2709 -0.4287 | 0.2634 -0.4011 | 0.3083 -0.4475 | 0.2989 -0.5602 |
| High Efficiency | 0.5825 -1.0000 | 0.6573 -1.0000 | 0.4746 -1.0000 | 0.4287 -1.0000 | 0.4011 -1.0000 | 0.4475 -1.0000 | 0.5602 -1.0000 |

## Analysis of factors influencing industrial restructuring in resource- based cities

The industrial transformation of resource-based cities can affect green land use efficiency through the injection and change of infrastructure construction, fixed asset investment, and other factors, especially the most significant impact on economic output benefits. To deeply understand the driving and restricting factors of urban land use from various aspects, Tobit regression is further used to analyze the factors affecting green land use efficiency.

In this paper, the P value of the Tobit regression test is 0. When the green use efficiency of land (LAND) is used as the explained variable, as shown in Table 7.

Industrial transformation (INS) is positive, and the influence coefficient of the industrial transformation on land use is 0.064 and significant at the level of 1%. This result shows that the proportion of urban construction land is positively correlated with industrial transformation, the industrial transformation of resource-based cities promotes rational land allocation and adjusts the extensive urban land use. Recently, with the upgrading of traditional production capacity and the optimization and adjustment of industrial structure in China, the spatial layout and utilization form from urban construction land has also changed. Based on this, the dispersed industrial land began to gather under the influence of industrial transformation, and several idle industrial and mining lands also appeared. Idle land and unreasonably used land make it impossible for industrial land will be integrated on a large scale, which also means that the use area of construction land is reducing.

Environmental regulation (ENVIRON) is positive, and the impact coefficient of environmental regulations on industrial restructuring was 8.251, which was significant at 1%. This indicates that environmental regulations play a facilitating role during the adjustment of industrial structures. The government protects the environment by making policies and publicizing the society, especially the emission of industrial gases and wastewater, which must motivate the relevant enterprises to adjust their production methods to save natural energy, and reduce emissions. As a result, the resource extraction industry, which is featured on high pollution and high emissions, has adjusted its production methods under the pressure of environmental regulations, and high efficiency and low consumption technologies oriented towards green development have been introduced into the research and development and production process, thus promoting the green use of land.

The regression result of government expenditure (GOVERN) shows a positive value and the coefficient of government expenditure on industrial restructuring is 128.7 and significant at a 1% level. This value indicates that government expenditure has a catalytic effect on the green use of land. On the one hand, the government introduces innovation support policies and increases innovation investment, bringing a good innovation environment for enterprises,

**Table 7. Results of Tobit regression.**

| Variable | Coef. | Std. Err. | T-statistic | P-value |
|---|---|---|---|---|
| INS | 0.064*** | 0.069 | 0.920 | 0.000 |
| ENVIRON | 8.251*** | 1.791 | 4.600 | 0.000 |
| PEOPLE | -345.170*** | 30.223 | -11.420 | 0.000 |
| GOVERN | 128.701*** | 14.474 | 8.890 | 0.000 |
| FDI | 32.830** | 1.721 | 0.220 | 0.035 |
| GREEN | 0.027* | 0.025 | 1.140 | 0.026 |
| _cons | 58.634 | 1.722 | 34.040 | 0.000 |

***,**,and* denote a significance level of 1%, 5%, and 10%, respectively. The corresponding t-values are in parentheses.

which can use the innovation results to adjust production methods and production processes. The upgrading of the industrial structure can promote technological innovation, which brings economic benefits to boost local fiscal revenue so that the government will continue to invest funds in industrial transformation and upgrading within a cycle. On the other hand, local governments can adjust the supply of public goods such as infrastructure construction or parks and greenery, and through fiscal instruments can urban economic structure change to a certain extent, thus causing changes in land use patterns and thus affecting the output of land green benefits.

The result of population density (PEOPLE) is negative, and the influence coefficient of population density on green land use is −345.17 and significant at the level of 1%. This result shows that population density inhibits green land use. With the rapid development of urbanization, the trend of incongruity between population growth, land use, and economic development has become serious. For example, the contradiction between man and land, the contradiction between supply and demand, and other problems hinder regional social and economic development. Especially in some emerging urban agglomerations, the interaction between economic development and population growth makes the contradiction between land use and population number more acute. On the one hand, the increase in population density has brought about a "demographic dividend." On the other hand, due to the influence of economic development and population agglomeration, the upgrading and streamlining of the secondary industry have a "siphon effect" that makes the population gather and play its role in spatial economic agglomeration.

The result of international investment (FDI) is positive, the influence coefficient of international investment on industrial transformation is 32.83, and the P value is 0.035, indicating that there is a significant difference. This result shows that foreign direct investment generates market competition and speeds up the flow of factors, thus promoting the development of the economy and industry. The attracted foreign investment can play a connecting role, driving the import and export of intermediate products, thus expanding the international market. For industrial transformation, OFDI can transfer overcapacity and inferior industries, which is conducive to optimizing the allocation of production factors and concentrating input factors in scientific and technological innovation industries, thus promoting the upgrading of industrial structure.

The results for green cover (GREEN) are positive, and the influence coefficient of green coverage in built-up areas on industrial transformation is 0.027, which is significant at the level of 10%. This result shows that urban greening construction has higher quality requirements for land-based input factors, which also means that environmental greening and land green cover utilization are coordinated with each other. The quality of the urban ecological environment requires the greening coverage of the city, and some enterprises with high pollution and high emissions have to change their production mode or suspend production.

## Discussion

We have evaluated the land-use efficiency of coal resource-based cities in China within the parameters of this study. Our research findings reveal that numerous factors influence and restrict the utilization of green spaces. In addition to the observations made by Xia et al. in 2020 [20], highlighting the impact of decision-making by land use stakeholders on the land use efficiency of coal resource-based cities. Furthermore, facing environmental pollution in resource-based cities, environmental regulation should be strengthened while industrial structure adjustment is carried out [68]. Our study posits that actively modifying the status of these influencing factors can enhance the efficiency of green space utilization.

While our investigation demonstrates a consistent spatio-temporal trend in urban land use efficiency, in line with the findings of Lu et al. in 2022 [19], it distinguishes itself by utilizing industrial transformation as the explanatory variable. This approach differs from the research framework focusing on industrial integration in urban land green use as explored in Lu et al. in 2018 [48]. Furthermore, our study extends its scope beyond a specific geographical basin, encompassing coal resource-based cities more broadly. In comparison to Zhu et al. in 2013 [22] study on specific resource-based cities, our research is more specialized, targeting the enhancement of land use efficiency in specific city types.

Due to the extensive scope of resource-based cities, this study solely focused on the land green use efficiency of coal resource-based cities and the impact of industrial transformation on land green use efficiency.The adjustment of input structures for land, labor, capital, and other factors, will impact the adjustment and alteration of the industrial structure's internal system. It provides development ideas for backward resource-based cities and untransformed resource-based cities. Considering the different resource statuses of resource-based cities based on different resources, it is necessary to carry out heterogeneity test. Therefore, future research will aim to collect more data on land use progress in other types of resource-based cities and investigate the influence of various factors on land green use efficiency. Additionally, this study did not consider the reciprocal effect of green land use on industrial transformation which may result in an analysis limited to influencing factors without a thorough exploration into the spillover effect that industrial transformation has on green land use efficiency.

Consequently, the evaluation of land green use efficiency should consider the additional effect of the industrial transformation process itself. By evaluating and analyzing coal resource-based cities' utilization efficiency for green spaces, inherent issues can be identified along with potential areas for improvement while guiding and promoting sustainable development and transforming resource-based cities. To be specific, the extensive scope of resource-based cities precluded the examination of other types in this study. Future research will gather more data on land use progress in different types of resource-based cities and investigate the influence of various factors on land green use efficiency. Considering the distinctive resource conditions found in other resource-based cities reliant on high energy consumption, pollution, and emission industries such as oil and metallurgy, it is essential to conduct heterogeneous studies to provide rich empirical evidence. Moreover, evaluations regarding land green use efficiency should also consider additional impacts during the process of industrial transformation.

## Conclusions

The evident shift towards green land use in resource-based cities holds significant importance for promoting industrial transformation and optimizing the economic development driving force. Overall, the SBM-Undesirable model presents a comprehensive assessment of land green use efficiency by considering multiple factors and mitigating the influence of undesirable outputs. This study examines the temporal and spatial evolution characteristics of land green use efficiency in China's coal resource-based cities and identifies the factors that enhance and optimize urban land green use efficiency via a regression analysis model. The calculation and in-depth analysis of land efficiency in coal resource-based cities reveal that: (1) The dynamics of land use indicate that China is undergoing a national land area and land use structure adjustment from 2014 to 2020. (2) The output-based assessment reveals that during the seven years from 2014 to 2020, the regions with high, medium, and low land use efficiency in China generally exhibit dynamic changes. As a factor of production, land is influenced by industrial restructuring. (3) The transformation of resource-based cities is a long-term construction

project. In future research, cities with low land use efficiency should focus on adjusting unreasonable land use methods and reconditioning backward industries and traditional economic development models. At the same time, paying attention to the transformation of land use mode and the optimization and upgrading of industrial structure are also the focus of future research on green land use efficiency. Along this path, cities could stimulating investment and economic development, and implementing the concepts of "the replacement of old growth drivers with new ones" and "green and sustainable development.

## Author Contributions

**Conceptualization:** Hongfeng Zhang.

**Data curation:** Yinuo Zhao.

**Formal analysis:** Yinuo Zhao.

**Funding acquisition:** Hongfeng Zhang.

**Investigation:** Yinuo Zhao.

**Methodology:** Hongfeng Zhang.

**Project administration:** Hongfeng Zhang.

**Supervision:** Yinuo Zhao, Fangteng Yang.

**Writing – review & editing:** Yinuo Zhao.

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
