## [Decision Letter · Decision Letter 0]

18 Dec 2023

PONE-D-23-38687Analyzing Green and Sustainable Land Use in China's Coal Cities: Insights from Industrial TransformationPLOS ONE

Dear Dr. Zhao,

Thank you for submitting your manuscript to PLOS ONE. After careful consideration, we feel that it has merit but does not fully meet PLOS ONE’s publication criteria as it currently stands. Therefore, we invite you to submit a revised version of the manuscript that addresses the points raised during the review process.

We look forward to receiving your revised manuscript.

Kind regards,

Rita Yi Man Li

Academic Editor

PLOS ONE

Journal Requirements:

"This research was funded by The Key Project of National Social Science Foundation of China (grant number 19AJY014); Major Project of Shandong Key Research and Development Program (Soft Science) (grant number 2023RZA02013); Shandong Province Social Science Planning Research Project (grant number 22CZTJ23); Provincial Applied Research Project of Humanities and Social Sciences of Shandong Province(2022-YYGL-08)."

"This research was funded by The Key Project of National Social Science Foundation of China (grant number 19AJY014); Major Project of Shandong Key Research and Development Program (Soft Science) (grant number 2023RZA02013); Shandong Province Social Science Planning Research Project (grant number 22CZTJ23); Provincial Applied Research Project of Humanities and Social Sciences of Shandong Province(2022-YYGL-08)."

"This research was funded by The Key Project of National Social Science Foundation of China (grant number 19AJY014); Major Project of Shandong Key Research and Development Program (Soft Science) (grant number 2023RZA02013); Shandong Province Social Science Planning Research Project (grant number 22CZTJ23); Provincial Applied Research Project of Humanities and Social Sciences of Shandong Province(2022-YYGL-08)."

6. We note that Figures 4 and S4 in your submission contain [map/satellite] images which may be copyrighted. All PLOS content is published under the Creative Commons Attribution License (CC BY 4.0), which means that the manuscript, images, and Supporting Information files will be freely available online, and any third party is permitted to access, download, copy, distribute, and use these materials in any way, even commercially, with proper attribution. For these reasons, we cannot publish previously copyrighted maps or satellite images created using proprietary data, such as Google software (Google Maps, Street View, and Earth). For more information, see our copyright guidelines: http://journals.plos.org/plosone/s/licenses-and-copyright.

a. You may seek permission from the original copyright holder of Figures 4 and S4 to publish the content specifically under the CC BY 4.0 license.  

7. We notice that your supplementary [figures/tables] are included in the manuscript file. Please remove them and upload them with the file type 'Supporting Information'. Please ensure that each Supporting Information file has a legend listed in the manuscript after the references list.

Additional Editor Comments :

Add research question in introduction.

Theoretical analysis framework, Influence mechanism of industrial restructuring in resource-based cities,F igure 1. The conduction of industrial transformation to land green utilization. Table 2. Variables associated with non-expected output models etc have missing citation.

Delete the title Research Methodology and add citation for this section.

Table 1 Energy consumption structure in coal-based areas (%). Why is this related to land use efficiency?

Figure 3. Dynamic changes of land use in resource-based cities cannot show the dynamics but differences among different places only.

Table 3. Explanatory variables associated with the non-expected output model factors calculation should be included.

What is land use efficiency? How do you define that?

State what have been done in land use in China recently: Comparative study of factors contributing to land surface temperature in high-density built environments in megacities using satellite imagery, Sustainability, 2021

Please add details about environmental regulation: Achieving Carbon Neutrality – The Role of Heterogeneous Environmental Regulations on Urban Green Innovation, Front. Ecol. Evol., Volume 10 - 2022

The impact of sustainability awareness and moral values on environmental laws- Sustainability, 2021

Extend conclusion, limitation What are the main contributions of the paper in terms of academic, policy and practical? What are the point of orginality?

The description of the authors have not covered the changes after the government policies of housing is for living but not speculation. The authors should provide insight after the new policies "Factors influencing large real estate companies' competitiveness: a sustainable development perspective, Land, 2021"

Reviewers' comments:

Reviewer's Responses to Questions

**Comments to the Author**

1. Is the manuscript technically sound, and do the data support the conclusions?

Reviewer #1: Yes

Reviewer #2: Yes

2. Has the statistical analysis been performed appropriately and rigorously? 

Reviewer #1: Yes

Reviewer #2: Yes

3. Have the authors made all data underlying the findings in their manuscript fully available?

Reviewer #1: Yes

Reviewer #2: Yes

4. Is the manuscript presented in an intelligible fashion and written in standard English?

Reviewer #1: Yes

Reviewer #2: Yes

5. Review Comments to the Author

Reviewer #1: The initial section briefly situates the study within a broader context; however, it fails to emphasize the significance of the research. Instead of merely mentioning the main purpose without acknowledging its importance, the introduction should clearly define the purpose of the work and its significance. Additionally, it is crucial to thoroughly review the current state of the research field and reference key publications. When necessary, controversial and diverging hypotheses should be highlighted. Lastly, the main aim of the work should be briefly mentioned, and the principal conclusions should be emphasized.

Authors are encouraged to engage in a comprehensive discussion of the outcomes, elucidating their potential interpretations in light of prior investigations and the underlying hypotheses. Furthermore, it is essential to deliberate upon the broader implications of the findings and their significance within a wider framework. Additionally, it is advisable to emphasize potential avenues for future research.

Reviewer #2: 1- Please read and check the paper thoroughly as there are sentences which has been found to be redundant (line 47)

2-To re-write the sentence from line 65-69 as they are rather confusing

3- Figure 1 (line 217) should be placed under the method section as it prematurely depicts the method used (the model and the applied regression test)

4-Suggestion to move the study limitations (line 559) to the early part of the paper and not under the discussion section.

6. PLOS authors have the option to publish the peer review history of their article (what does this mean?). If published, this will include your full peer review and any attached files.

Reviewer #1: **Yes: **Mohamed A. E. AbdelRahman

Reviewer #2: No

---

## [Author Response · Author response to Decision Letter 0]

9 Jan 2024

Response to Reviewer #1 

Dear Reviewer #1 :

Thank you for your letter and comments concerning our manuscript entitled.

We sincerely thank the editor and all reviewers for their valuable feed-back that we have used to improve the quality of our manuscript. The reviewer comments are laid out below in italicized font and specific concerns have been numbered. Our response is given in normal font and changes/additions to the manuscript are given in the red text.

Reviewer #1 1- The initial section briefly situates the study within a broader context; however, it fails to emphasize the significance of the research. Instead of merely mentioning the main purpose without acknowledging its importance, the introduction should clearly define the purpose of the work and its significance.

Author’s reply :

We sincerely thank the reviewer for careful reading. We appreciate the constructive feedback, and, in response, in the INTRODUCTION, we make clear the research background and repeatedly emphasize the research purpose of this research. Please see the specific modification in line 29-31, 71-74, 91-93.

Reviewer #1 2- Additionally, it is crucial to thoroughly review the current state of the research field and reference key publications. When necessary, controversial and diverging hypotheses should be highlighted.

Author’s reply :

We acknowledge the reviewers' observations regarding the literature review in our article and recognize the identified issues of incompleteness, inadequacy, and a lack of objectivity. We appreciate the constructive feedback, and, in response, we have diligently worked to reorganize and refine the literature review section. The revisions aim to enhance the completeness and objectivity of our analysis. We thank the reviewers for guiding us toward improvements and apologize for any inconvenience caused by the initial shortcomings. 

Reviewer #1 3- Lastly, the main aim of the work should be briefly mentioned, and the principal conclusions should be emphasized.

Author’s reply :

We think this is an excellent suggestion, and, in response, In the CONCLUSION section, we have comprehensively reviewed the work done in this paper on the analysis of green land use efficiency in coal resource-based cities, and list the research results one by one. The details are as follows:

The evident shift towards green land use in resource-based cities holds significant importance for promoting industrial transformation and optimizing the economic development driving force. Overall, the SBM-Undesirable model presents a comprehensive assessment of land green use efficiency by considering multiple factors and mitigating the influence of undesirable outputs (Main tasks). This study examines the temporal and spatial evolution characteristics of land green use efficiency in China's coal resource-based cities and identifies the factors that enhance and optimize urban land green use efficiency via a regression analysis model (Purpose of the research). The calculation and in-depth analysis of land efficiency in coal resource-based cities reveal that: (1) The dynamics of land use indicate that China is undergoing a national land area and land use structure adjustment from 2014 to 2020. (2) The output-based assessment reveals that during the seven years from 2014 to 2020, the regions with high, medium, and low land use efficiency in China generally exhibit dynamic changes. As a factor of production, land is influenced by industrial restructuring. (3) The transformation of resource-based cities is a long-term construction project. The adjustment of input structures for land, labor, capital, and other factors, as well as the shift in traditional production concepts, will impact the adjustment and alteration of the industrial structure's internal system (Conclusions of the research). Low land use efficiency cities should focus on adjusting inappropriate land use patterns, rectifying backward industries and traditional economic development models, promoting the transformation of land use methods and the optimization and upgrading of industrial structures, stimulating investment and economic development, and implementing the concepts of "the replacement of old growth drivers with new ones" and "green and sustainable development (Research Recommendations).

Reviewer #1 4- Authors are encouraged to engage in a comprehensive discussion of the outcomes, elucidating their potential interpretations in light of prior investigations and the underlying hypotheses. Furthermore, it is essential to deliberate upon the broader implications of the findings and their significance within a wider framework. Additionally, it is advisable to emphasize potential avenues for future research.

Author’s reply :

We acknowledge the reviewers' observations regarding the discussion and conclusion in our article and recognize the identified issues of incompleteness, inadequacy. We appreciate the constructive feedback, and, in response, we have diligently worked to reorganize and refine the discussion and conclusion. The revisions aim to enhance the completeness of our analysis. We thank the reviewers for guiding us toward improvements and apologize for any inconvenience caused by the initial shortcomings. The details are as follows:

Due to the extensive scope of resource-based cities, this study solely focused on the land green use efficiency of coal resource-based cities and the impact of industrial transformation on land green use efficiency. It provides development ideas for backward resource-based cities and untransformed resource-based cities. Considering the different resource statuses of resource-based cities based on different resources, it is necessary to carry out heterogeneity test. Therefore, future research will aim to collect more data on land use progress in other types of resource-based cities and investigate the influence of various factors on land green use efficiency. Additionally, this study did not consider the reciprocal effect of green land use on industrial transformation which may result in an analysis limited to influencing factors without a thorough exploration into the spillover effect that industrial transformation has on green land use efficiency.

Consequently, the evaluation of land green use efficiency should consider the additional effect of the industrial transformation process itself. By evaluating and analyzing coal resource-based cities' utilization efficiency for green spaces, inherent issues can be identified along with potential areas for improvement while guiding and promoting sustainable development and transforming resource-based cities. To be specific, the extensive scope of resource-based cities precluded the examination of other types in this study. Future research will gather more data on land use progress in different types of resource-based cities and investigate the influence of various factors on land green use efficiency. Considering the distinctive resource conditions found in other resource-based cities reliant on high energy consumption, pollution, and emission industries such as oil and metallurgy, it is essential to conduct heterogeneous studies to provide rich empirical evidence. Moreover, evaluations regarding land green use efficiency should also consider additional impacts during the process of industrial transformation.

Thanks again to the reviewers and editors for their valuable comments and suggestions on this article, which further improves the quality of the paper! We have learned from your valuable comments to improve the level of article writing experience and the spirit of excellence. If there is anything wrong with this article, please inform us, we will continue to improve and modify, thank you very much!

Response to Reviewer #2

Dear Reviewer #2 :

Thank you for your letter and comments concerning our manuscript entitled

We sincerely thank the editor and all reviewers for their valuable feed-back that we have used to improve the quality of our manuscript. The reviewer comments are laid out below in italicized font and specific concerns have been numbered. Our response is given in normal font and changes/additions to the manuscript are given in the red text.

Reviewer #2 1- Please read and check the paper thoroughly as there are sentences which has been found to be redundant (line 47).

Author’s reply

We sincerely thank the reviewer for careful reading. As suggested by the reviewer, we have rechecked the redundant sentences and rewritten the section (line 41-42).

Reviewer #2 2-To re-write the sentence from line 65-69 as they are rather confusing.

Author’s reply

We feel sorry for our carelessness. In our resubmitted manuscript, We have re-written this part according to the Reviewer's suggestion (line 71-77). Thanks for your correction.

Reviewer #2 3- Figure 1 (line 217) should be placed under the method section as it prematurely depicts the method used (the model and the applied regression test).

Author’s reply

We think this is an excellent suggestion. We have adjusted the position of Figure 1, the new position is after the research method Tobit regression section in the revised version (line 335-336).

Reviewer #2 4-Suggestion to move the study limitations (line 559) to the early part of the paper and not under the discussion section.

Author’s reply

We think this is an excellent suggestion, we appreciate the constructive feedback. The revisions aim to enhance the completeness of our analysis, the research limitations are briefly presented in the INTRODUCTION section of the research and expanded again in the DISCUSSION, in order to point out the research limitations and potential avenues for future research in detail.

Thanks again to the reviewers and editors for their valuable comments and suggestions on this article, which further improves the quality of the paper! We have learned from your valuable comments to improve the level of article writing experience and the spirit of excellence. If there is anything wrong with this article, please inform us, we will continue to improve and modify, thank you very much!

Response to editor

Dear Editor:

Thank you for your letter and comments concerning our manuscript entitled (Manuscript ID: PONE-D-23-38687).

Those comments are all valuable and very helpful for revising and improving our research. According to the editor comments. we have made extensive modifications to our manuscript. In this revised version. The reviewer comments are laid out below in italicized font and specific concerns have been numbered. Our response is given in normal font and changes/additions to the manuscript are given in the red text. the detailed corrections are listed below.

According to the requirements section of the journal, we have made corresponding modifications, including：

Journal Requirements 1. Please ensure that your manuscript meets PLOS ONE's style requirements, including those for file naming.

Author’s reply

Thanks to the editor for your valuable comments on the article style specification, we fully agree with your suggestions. And according to PLOSOne_formatting_sample_main_body, the full article is adjusted.

Journal Requirements 2. PLOS requires an ORCID iD for the corresponding author in Editorial Manager on papers submitted after December 6th, 2016. Please ensure that you have an ORCID iD and that it is validated in Editorial Manager. To do this, go to ‘Update my Information’ (in the upper left-hand corner of the main menu), and click on the Fetch/Validate link next to the ORCID field.

Author’s reply

Thanks to the editor for your valuable comments on the article, we fully agree with your suggestions. We have verified the corresponding author's ORCID iD as required.

Journal Requirements 4. Please state what role the funders took in the study.

Author’s reply

Thanks to the editor for your valuable comments on the article, we fully agree with your suggestions. We state: "The funders had no role in study design, data collection and analysis, decision to publish, or preparation of the manuscript." The amended role statement is attached。

Journal Requirements 5. Please remove any funding-related text from the manuscript and let us know how you would like to update your Funding Statement.

Author’s reply

Thanks to the editor for your valuable comments on the article, we fully agree with your suggestions. We have deleted all the words related to funding in the manuscript.

Journal Requirements 6. We require you to either (1) present written permission from the copyright holder to publish these figures specifically under the CC BY 4.0 license, or (2) remove the figures from your submission.

Author’s reply

Thanks to the editor for your valuable comments on the article, we fully agree with your suggestions.

Since the original copyright of Figure 4 belongs to the National Platform for Common Geospatial Information Services, it is difficult to obtain written permission from the copyright owner. Therefore, we change the presentation form of Figure 4 to a data statistics table with a natural breakpoint table attached. To facilitate the understanding of reviewers, editors and readers, and make corresponding corrections in the supporting information file.

According to the section of Additional Editor Comments, we have made corresponding modifications, including：

Editor Comments 1. Add research question in introduction.

Author’s reply

Thanks to the editor for your valuable comments on the introduction, we fully agree with your suggestions. The research questions of this article have been added in line 29-31, and 71-74 of the introduction.

Editor Comments 2. Heoretical analysis framework, Influence mechanism of industrial restructuring in resource-based cities,F igure 1. The conduction of industrial transformation to land green utilization. Table 2. Variables associated with non-expected output models etc have missing citation.

Author’s reply

Thanks to the editor for your valuable comments on the figures and tables, We have cross-referenced the citations of the indicators in Table 2 Variables associated with non-expected output models. However, Figure 1 The conduction of industrial transformation to land green utilization is summarized by us according to the research track and research topic of the article, without reference to any literature, so it is not recommended to quote. Thank you for your normative reminder.

Editor Comments 3. Table 1 Energy consumption structure in coal-based areas (%). Why is this related to land use efficiency?

Author’s reply

Thanks to the editor for your valuable comments.For the design of Table 1, our idea is to highlight coal as the main energy of coal resource-based cities, so as to distinguish them from other resource-based cities, and to show that coal as the main energy of coal resource-based cities will continue to be the economic driving force and the focus of urban transformation of coal resource-based cities in the future.

Editor Comments 4. Figure 3. Dynamic changes of land use in resource-based cities cannot show the dynamics but differences among different places only.

Author’s reply

We appreciate the constructive feedback, and, in response, we change Figure 3 to a schematic diagram that includes a combination of time, place name and range of variation. The revisions aim to enhance the completeness of our analysis.

Editor Comments 7. What is land use efficiency? How do you define that?

Author’s reply

Thanks to the editor for your valuable comments on the contents of article. By summarizing the domestic and foreign references, we summarized the land use efficiency as land use efficiency refers to the efficiency level of economic, social and ecological benefits generated by the allocation and use of land resources in different regions and economic sectors, and rewrote it in line 144-146 of the manuscript.

Editor Comments 8.State what have been done in land use in China recently.

Author’s reply

We sincerely appreciate the valuable comments. The revisions aim to enhance the completeness of our analysis. We have checked the literature carefully and added more references on and into the THEORETICAL ANALYSIS FRAMEWORK Review of the literature part in the revised manuscript. As suggested by the reviewer, we have added more references to su

---

## [Decision Letter · Decision Letter 1]

27 Feb 2024

PONE-D-23-38687R1Analyzing Green and Sustainable Land Use in China's Coal Cities: Insights from Industrial TransformationPLOS ONE

Dear Dr. Zhao,

Thank you for submitting your manuscript to PLOS ONE. After careful consideration, we feel that it has merit but does not fully meet PLOS ONE’s publication criteria as it currently stands. Therefore, we invite you to submit a revised version of the manuscript that addresses the points raised during the review process.

We look forward to receiving your revised manuscript.

Kind regards,

Rita Yi Man Li

Academic Editor

PLOS ONE

Journal Requirements:

Additional Editor Comments:

Add citation for Theoretical analysis framework.

Table 1 heading should be a new row?

The transformation of resource-based cities is progressing at a rapid pace, Materials and Data, Land use dynamic Model need citation.

Spatial temporal need citation and explain what is that Uncovering PM2. 5 transport trajectories and sources at district within city scale, Journal of Cleaner Production, 2023

Many sections still have missing citations. After adding all, I hope to see it publish.

Reviewers' comments:

Reviewer's Responses to Questions

**Comments to the Author**

1. If the authors have adequately addressed your comments raised in a previous round of review and you feel that this manuscript is now acceptable for publication, you may indicate that here to bypass the “Comments to the Author” section, enter your conflict of interest statement in the “Confidential to Editor” section, and submit your "Accept" recommendation.

Reviewer #1: All comments have been addressed

Reviewer #3: All comments have been addressed

2. Is the manuscript technically sound, and do the data support the conclusions?

Reviewer #1: Yes

Reviewer #3: No

3. Has the statistical analysis been performed appropriately and rigorously? 

Reviewer #1: No

Reviewer #3: Yes

4. Have the authors made all data underlying the findings in their manuscript fully available?

Reviewer #1: Yes

Reviewer #3: Yes

5. Is the manuscript presented in an intelligible fashion and written in standard English?

Reviewer #1: Yes

Reviewer #3: No

6. Review Comments to the Author

Reviewer #1: The authors have adequately addressed your comments raised in a previous round of review and I feel that this manuscript is now acceptable for publicationThe manuscript maybe accepted for publication

Reviewer #3: I limit my comments and questions below to the issues raised in my initial review. However, to genuinely contribute with an article that is expected to synthesize previous contributions in the field and point out further Research, i deem that the authors need to do some more work, particularly to sharpen the current manuscript. - Literature Review has the chance to be further improved: it seems that the authors have made the retrospection. However, via the review, what issues should be addressed? What implication can be referred to?

- In the last paragraph must be done a summary (resume) of the paper, i.e., a clear idea about

what will be studied in the paper. The last paragraph's purpose in the Introduction section is to summarize the main points, restate the paper's main idea, and show how the paper statements were proven. It should have the general objective of the work that the authors must write

-Regarding conclusions, they are the most important output of the Research. Therefore, it is suggested that the findings be explored more. Implications for future research may also be included in the conclusion at the end.

-Some sentences from the conclusion could be moved up in the discussion section. A conclusion section must be well written and clearly explain the study findings.

- Implications for future research may also be included in the conclusion. This Research has article has created a lively discussion on so many issues that were hitherto unheard of and not addressed.

-Check the citations and references (one by one) if there is any missing information. Citations and references must be 100% accurate according to the journal guidelines.

7. PLOS authors have the option to publish the peer review history of their article (what does this mean?). If published, this will include your full peer review and any attached files.

Reviewer #1: **Yes: **Mohamed A. E. AbdelRahman

Reviewer #3: No

---

## [Author Response · Author response to Decision Letter 1]

12 Mar 2024

Response to Reviewer #1

Dear Reviewer #1 :

We would like to thank you for your professional review work, constructive comments, and valuable suggestions on our manuscript.

We wish you all the best in your work!

Response to Reviewer #3

Dear Reviewer #3 :

Thank you for your letter and comments concerning our manuscript entitled

We sincerely thank the editor and all reviewers for their valuable feed-back that we have used to improve the quality of our manuscript. The reviewer comments are laid out below in italicized font and specific concerns have been numbered. Our response is given in normal font and changes/additions to the manuscript are given in the red text. At the same time, we have recruited native English speakers to polish and revise the manuscript.

Reviewer #3 1- Literature Review has the chance to be further improved: it seems that the authors have made the retrospection. However, via the review, what issues should be addressed? What implication can be referred to?

Author’s reply

We acknowledge the reviewers' observations regarding the literature review in our article and recognize the identified issues of incompleteness, inadequacy, and a lack of objectivity. We appreciate the constructive feedback, and, in response, in the last part of the literature review, we summarize the existing problems in the industrial structure and urban transformation, hoping to find a way to solve the green use of resource-based urban land according to the existing problems(line 222-228).

Reviewer #3 2-In the last paragraph must be done a summary (resume) of the paper, i.e., a clear idea about what will be studied in the paper. The last paragraph's purpose in the Introduction section is to summarize the main points, restate the paper's main idea, and show how the paper statements were proven. It should have the general objective of the work that the authors must write

Author’s reply

We sincerely thank the reviewer for careful reading. We appreciate the constructive feedback, in response, in the INTRODUCTION, On the basis of the existing research purpose, we add the purpose of this paper to solve the constraints (line 96-97).

Reviewer #3 3&5- Regarding conclusions, they are the most important output of the Research. Therefore, it is suggested that the findings be explored more. Implications for future research may also be included in the conclusion at the end.

Author’s reply

We think this is an excellent suggestion, and, in response, In the CONCLUSION section, In our resubmitted manuscript, in the last part of the conclusion, we review the work objectives and research conclusions of the whole paper, summarize the future research direction, and make an outlook on the future development of green land use combined with industrial economy and other factors. (line 642-649). Thanks for your correction.

Reviewer #3 4-Some sentences from the conclusion could be moved up in the discussion section. A conclusion section must be well written and clearly explain the study findings.

Author’s reply

We think this is an excellent suggestion, we appreciate the constructive feedback. We have adjusted some statements of the discussion and conclusion, enriched the results of our work, and introduced the limitations of our research, hoping to provide readers with ideas of existing research and directions for future research in the discussion section(line 605-606).

Reviewer #3 6-Check the citations and references (one by one) if there is any missing information. Citations and references must be 100% accurate according to the journal guidelines.

Author’s reply

We sincerely thank the reviewer for careful reading. As suggested by the reviewer, we have carefully rechecked the citations and references. After checking all the citations, the format and related information of the references were adjusted and improved according to the journal guidelines

Thanks again to the reviewers and editors for their valuable comments and suggestions on this article, which further improves the quality of the paper! We have learned from your valuable comments to improve the level of article writing experience and the spirit of excellence. If there is anything wrong with this article, please inform us, we will continue to improve and modify, thank you very much!

---

## [Decision Letter · Decision Letter 2]

25 Mar 2024

Analyzing Green and Sustainable Land Use in China's Coal Cities: Insights from Industrial Transformation

PONE-D-23-38687R2

Dear authors,

We’re pleased to inform you that your manuscript has been judged scientifically suitable for publication and will be formally accepted for publication once it meets all outstanding technical requirements.

Kind regards,

Rita Yi Man Li

Academic Editor

PLOS ONE

Additional Editor Comments (optional):

Reviewers' comments:

Reviewer's Responses to Questions

**Comments to the Author**

1. If the authors have adequately addressed your comments raised in a previous round of review and you feel that this manuscript is now acceptable for publication, you may indicate that here to bypass the “Comments to the Author” section, enter your conflict of interest statement in the “Confidential to Editor” section, and submit your "Accept" recommendation.

Reviewer #1: All comments have been addressed

Reviewer #4: All comments have been addressed

2. Is the manuscript technically sound, and do the data support the conclusions?

Reviewer #1: No

Reviewer #4: Yes

3. Has the statistical analysis been performed appropriately and rigorously? 

Reviewer #1: No

Reviewer #4: Yes

4. Have the authors made all data underlying the findings in their manuscript fully available?

Reviewer #1: Yes

Reviewer #4: Yes

5. Is the manuscript presented in an intelligible fashion and written in standard English?

Reviewer #1: Yes

Reviewer #4: (No Response)

6. Review Comments to the Author

Reviewer #1: the authors have adequately addressed your comments raised in a previous round of review and i feel that this manuscript is now acceptable for publication

Reviewer #4: The study presents the results of original research.

In the article titled "Analyzing Green and Sustainable Land Use in China's Coal Cities: Insights from Industrial Transformation", it is aimed to investigate the spatial-temporal changes of industrial transformation in resource-based cities and its impact on the efficiency of green space use.

The study provides valuable information for policymakers and urban planners looking to promote sustainable development in resource-based cities.

7. PLOS authors have the option to publish the peer review history of their article (what does this mean?). If published, this will include your full peer review and any attached files.

Reviewer #1: **Yes: **Mohamed A. E. AbdelRahman

Reviewer #4: No
